# Insights into Human-Induced Pluripotent Stem Cell-Derived Astrocytes in Neurodegenerative Disorders

**DOI:** 10.3390/biom12030344

**Published:** 2022-02-23

**Authors:** Mandeep Kumar, Nhung Thi Phuong Nguyen, Marco Milanese, Giambattista Bonanno

**Affiliations:** 1Department of Pharmacy, Unit of Pharmacology and Toxicology, University of Genoa, 16148 Genoa, Italy; mandeep.pharm@gmail.com (M.K.); phuongnhungqt@gmail.com (N.T.P.N.); milanese@difar.unige.it (M.M.); 2Inter-University Center for the Promotion of the 3Rs Principles in Teaching & Research (Centro 3R), 16148 Genoa, Italy; 3IRCCS Ospedale Policlinico San Martino, 16132 Genoa, Italy

**Keywords:** human-induced pluripotent stem cells, neurodegenerative disease, iPSC-derived astrocytes, disease modeling

## Abstract

Most neurodegenerative disorders have complex and still unresolved pathology characterized by progressive neuronal damage and death. Astrocytes, the most-abundant non-neuronal cell population in the central nervous system, play a vital role in these processes. They are involved in various functions in the brain, such as the regulation of synapse formation, neuroinflammation, and lactate and glutamate levels. The development of human-induced pluripotent stem cells (iPSCs) reformed the research in neurodegenerative disorders allowing for the generation of disease-relevant neuronal and non-neuronal cell types that can help in disease modeling, drug screening, and, possibly, cell transplantation strategies. In the last 14 years, the differentiation of human iPSCs into astrocytes allowed for the opportunity to explore the contribution of astrocytes to neurodegenerative diseases. This review discusses the development protocols and applications of human iPSC-derived astrocytes in the most common neurodegenerative conditions.

## 1. Introduction

Neurodegenerative diseases involve chronic and progressive damage to neurons, resulting in various complications such as cognitive impairment, memory deficits, changes in perception and mood, and loss of sensitivity and motor abilities [1]. Most neurodegenerative diseases share common pathological pathways such as the over-accumulation of toxic-aggregated proteins, mitochondrial dysfunction, axonal transport defects, excitotoxicity, and chronic inflammation, which lead to neurodegeneration [2]. The increasing incidence of neurodegenerative disorders such as Alzheimer’s disease (AD) [3], amyotrophic lateral sclerosis (ALS) [4], multiple sclerosis (MS) [5], and Parkinson’s disease (PD) [6] represents a significant impact on global health because of the severity of their symptoms and the lack of effective therapies [7].

Due to the complexity of the pathological disease mechanisms and the challenge of accessing human samples, human modeling of neurodegenerative diseases has always been a difficult task. Animal models have commonly been used, due to the intrinsic features of in vivo preclinical experimentation and the possibility to establish ex vivo mechanistic studies. However, animal model studies usually require many animals and take a long time, thus limiting the number of experimental conditions tested [8]. The development of induced pluripotent stem cell (iPSC) allowed the creation of a new scenario, complementing animal disease models of neurodegenerative diseases. 

In 2007, Yamanaka and colleagues paved the way for this achievement by describing iPSC technology implemented in mice [9]. They generate iPSCs from embryonic fibroblasts and adult tail-tip fibroblasts of mice using four transcription factors: Octamer-binding transcription factor 3/4 (Oct3/4), sex-determining region Y (Sox2), the master regulator of cell cycle entry and proliferative metabolism (c-Myc), and Krüppel-like factor 4 (Klf4). This protocol took almost 30 days to generate rodent iPSCs and a further 30 days to differentiate them [9]. Subsequent studies showed that human iPSCs could also differentiate into any cell type in the human body; thus, well-established models of human disease, including both developmental and adult-onset diseases, often use iPSCs from patients [10,11]. Due to the rapid growth of discoveries, it is now possible to differentiate iPSCs from patients and healthy donors in almost all neural cells, allowing human neurodegenerative disease modeling with intrinsic advantages over in-vivo models. Advances in iPSC technology are fundamental to adapt astrocyte research to translational studies since human astrocytes derived from fibroblasts of patients match phenotypes of various neurodegenerative diseases such as AD [12,13], ALS [14,15], HD [16], and fragile X syndrome [17]. Nevertheless, establishing iPSC-derived astrocytes as bona fide models of human astrocytes and individual heterogeneity of in vivo astrocytes should be approached with caution since these cells may modify the phenotype according to the culture conditions [18]. This review discusses human iPSC-derived astrocytes’ development protocols and applications in the most common neurodegenerative conditions.

## 2. Astrocytes Phenotypes and Functions

Astrocytes, the most abundant glial cell type in the central nervous system (CNS), play an essential role in neuronal development and function and are very specialized and heterogeneous throughout the CNS. They help maintain synaptogenesis, synaptic plasticity, the extracellular ion concentration, the blood–brain barrier (BBB), promote myelination in the white matter, and support neurons [19]. Apart from their fundamental physiological functions, it is generally agreed that astrocytes play a role in toxic/pathological events and that their dysfunction can generate neurological disorders such as neurodegenerative diseases, neurodevelopmental diseases, epilepsy, and astrogliomas [20]. In classical taxonomy, protoplasmic and fibrous astrocytes are two prominent gray and white matter classes, respectively [21]. Astrocytes form large homologous networks linked by gap junctions and heterologous networks with oligodendrocytes in different brain regions [22,23]. Recent studies showed that astrocytes change their morphology and function depending on age and their location in the brain [24,25,26,27]. 

Reactive gliosis is defined as changes in the neuroglia morphology, representing an actual reaction rather than a simple indicator of noxious influences on the CNS. Accordingly, the terms “reactive astrogliosis” and “astrocyte reactivity” define a pathological event that determines morphological, biochemical, and metabolic remodeling other than transcriptional regulation, which translates into the gain or loss of homeostatic functions [28]. Thus, morphology does not represent the only way to depict the role of astrocytes in neuropathologies since it does not correlate with functional phenotypes or their ability to impact the biology of other cells. Overall, the impact of reactive astrocytes varies and is complex and may depend on the pathological condition, which can be beneficial in one disease and detrimental in another [29]. The scenario is even more complex due to the presence of reactive microglia. Microglia can shift astrocyte signaling from a physiological to pathological status by increasing the production of factors such as tumor necrosis factor α (TNFα), which in turn has been reported to alter synaptic functions and behavior [30].

Specific molecular profiles and functions characterize the reactivity of astrocytes and their distinct impact on diseases and produce astrocyte phenotypes, representing the unique outcome of a given state. Notably, astrocyte reactivity characterizes pathological contexts while astrocyte activation represents physiological conditions [31]. Astrocyte reactivity is secondary to extrinsic signals and may evolve differently, exhibiting pathological modification and etiological features with a remarkable impact on disease progression [32].

In recent decades, scientists dramatically accelerated their research and deepened the roles of astrocytes, thanks to new interdisciplinary approaches combining omics with physiology and genetics. In this context, transcriptomics and proteomics have shown that in the healthy brain, following embryonic patterning programs or local neuronal cues, astrocytes are specialized to perform specific tasks in distinct CNS circuits [33]. The advent of genetic engineering opened a new research phase based on astrocyte-targeted manipulation, thus extending studies on detrimental astrocyte phenotypes. These studies identified other functional alterations in reactive astrocytes involving neurotransmitter and ion buffering, gap junctions, phagocytic activity, metabolic coupling, and most importantly, increased neuronal death due to the release of toxic factors and not merely the loss of trophic or antioxidant support from astrocytes [30].

This evolution generated the need to unify the nomenclature and refine the concepts that precisely define astrocyte heterogeneity by using a systematic approach to contextualize the contribution of astrocytes to CNS disorders. Recent studies have proposed that mouse astrocytes have two discrete pathological conditions: The A1 neurotoxic phenotype when exposed to specific cytokines secreted by activated microglia and the A2 neuroprotective phenotype when exposed to an experimental paradigm mimicking ischemic stroke [34,35]. However, this binary polarization should be approached cautiously since more recent studies recommend moving beyond the A1–A2 classification [30]. Indeed, only a subset of mixed A1 and A2 or pan-reactive transcripts are up-regulated in astrocytes of human patients or mouse models of chronic diseases or acute CNS injuries [36,37,38,39]. Moreover, advanced multidimensional data and the co-clustering statistical approach revealed that the A1 and A2 transcriptomes represent only two of the many potential astrocyte transcriptomes. As transcriptomic data are a fundamental tool used to characterize the functional diversity of reactive astrocytes, a multidimensional analysis should be applied to establish the uniqueness of astrocyte phenotypes [40].

## 3. Preparation of Human iPSCs-Derived Astrocytes

The literature shows significant differences between human and rodent astrocytes [25,41]. Human cortical astrocytes are larger, more complex, and more diverse. They present a diameter that is about 2.6-fold greater and express 10-fold more glial fibrillary acidic protein (GFAP) than rodent astrocytes. Moreover, human astrocytes propagate four times faster in Ca^2+^ waves than rodent astrocytes [41]. This evidence supports the need to create new in vitro models for studying neurodegenerative and neurologic diseases besides the widely used rodent models. 

Another successful outcome in the field by the same group that set up the induction of pluripotent stem cells from rodent fibroblast cultures was the development of a technique able to modify human fibroblasts to embryonic pluripotent undifferentiated cells and directly differentiate them into mature cells, such as astrocytes [11]. 

Since the discovery of the iPSC technique, many studies have followed the original method and generated iPSCs from human biopsy samples by recapturing the embryonic developmental status before differentiation into astrocytes [42,43]. The general mechanism of producing astrocytes from iPSCs is mainly based on four steps [44,45]:Formation of rosette-forming neuroepithelial cells from iPSCs;Generation of the neural stem cell;Expansion of induced neural stem cells in suspension culture with growth factors;Astrocyte differentiation and maturation.

Figure 1 schematizes this four-step process.

The protocols have been continuously upgraded to improve efficiency and functionality. They differ significantly in multiple aspects such as the cell seeding density at plating, substrates, media composition, timing and concentration of exogenous growth factors and morphogens used, and physical dimensions of the culture system. This section examines some of the commonly used techniques in astrocyte differentiation to provide a point of reference. Table 1 summarizes the most relevant procedures reported in the literature.

In differentiating iPSCs into astrocytes, most studies used combinations of morphogens and growth factors such as retinoic acid, the wingless-type MMTV integration site protein family, fibroblast growth factor (FGF), and the growth differentiation factor [9,47,59,60]. Region-specific astrocyte subtypes, including the ventral spinal cord, midbrain, and neocortical astrocytes, were generated using other supplements [61,62]. Roybon and colleagues generated spinal cord astrocytes from human iPSCs by using FGF1 and FGF2 [61]. In another study, Staffan and colleagues generated astrocytes from ventral midbrain and ventral spinal cord neural progenitors by using N2, B27, LDN-193189, SB-431542, Sonic Hedgehog (SHH-C25II) N-terminus, CHIR-99021, and 1 μM of smoothened agonist (SAG) [62]. 

Previously, neuroglial studies commonly used tissue culture techniques leading to insufficient cell growth. Serum-supplemented media containing unknown amounts of growth factors and hormones that could potentially alter or mask the action of other compounds have been introduced to overcome this issue. To eliminate the presence of extra, usually unidentified factors, serum-free and chemically defined (CD) media were developed [18]. These studies used a CD medium for primary astrocytes containing growth factors such as putrescine, prostaglandin Fz, insulin, fibroblast growth factor, and hydrocortisone, which induce dramatic changes in the morphological characteristics of cultured astrocytes compared to cells grown in a serum-free or serum-supplemented medium [63,64] and modifications of iPSCs-derived astrocyte phenotype [18]. 

Further development of the technology allowed the use of iPSCs in combination with gene editing and a three-dimensional (3D) organoid [9,65]. A study conducted by Sloan and colleagues in 2017 [66] showed that long-term cultivation in a 3D structure could stimulate the generation of iPSC-derived astrocytes. These astrocytes are closely related to primary human fetal astrocytes, and over time, in vitro, their gene-expression patterns and functions resemble those of mature human astrocytes. Astrocytes obtained from iPSCs in 3D cultures partially summarize functional networks as assessed by calcium influx spread [66,67]. It will be challenging to build a culture system consisting of neuronal or non-neuronal 3D cell co-cultures that mimic the human brain to model neurological and neurodegenerative disorders. 

Apart from the intrinsic features of the above methods, they are time-consuming and have low efficiency. The initial protocols took up to 180 days to isolate the iPSCs and differentiate them into astrocytes [47,48,67]. In this context, scientists have attempted to develop more-rapid methods to obtain neural cells with characteristics like those described above. One study directly converted skin fibroblasts into astrocytes by using various transcription factors such as nuclear factor IA (NFIA), NFIB, and sex-determining region (SRY)-box transcription factor 9 (SOX9), producing an astrocytic phenotype in human fibroblasts [44]. Two other studies demonstrated that human iPSCs acquire glial fate by activating NFIA and SOX9 pathways and further differentiate into functional astrocytes [57,68]. By using NFIA or NFIA and SOX9, they successfully produced astrocytes in 4–7 weeks. Taking advantage of the technical progress, recent protocols produced functional astrocytes in 30 days [51,69] and, using SOX9 and NFIB as transcription factors, even in 2 weeks [56].

Another recent study described a rapid, highly reproducible method to obtain astrocytes from human skin fibroblast using a different approach. The Authors directly converted human skin fibroblasts into inducible neuronal progenitor cells (iNPCs) in 18 days by transfecting a mixture of retroviral-expressing Klf4, Oct-3/4, Sox2, and c-Myc1 [54]. The generation of induced neural progenitor cell (iNPC)-derived astrocytes represents an up-and-coming alternative to iPSC-derived astrocytes. They are a renewable, readily available resource of human glia able to retain the age-related features of donor fibroblasts, thus making them a valuable model for interrogating astrocyte function over time in human CNS health and disease [54,58]. The authors used this method to model neurodegenerative diseases such as ALS, which allows the study of ALS while the patient is still alive to test potential therapeutics [70,71]. 

Exploring the molecular mechanisms leading to astrocyte differentiation will undoubtedly lead, in the future, to enhanced procedures to obtain iPSC-derived astrocytes with better efficiency and improved consistency with the human in situ astrocytes. Figure 2 shows a schematic representation of how to obtain iPSCs and iNPCs from somatic cells and derive astrocytes relevant to neurodegenerative disorders from them.

iPSC-derived astrocytes have reached the stage where we can consider them as a tool to solve various research questions and even for clinical applications. It is now important to take stock, identify remaining research needs, and work toward protocols that provide standardized sets of cells. To our knowledge, none of the available protocols are free of limitations, which are as follows: They provide insufficient details for reproduction;Astrocyte purity and maturity are not comparable to those obtained with primary astrocyte cultures;There are technical issues that prohibit widespread use, even though the protocol may be suitable;Emerging astrocytes are often poorly characterized.

For these reasons, no present protocol has become a gold standard, and there is still a need for further protocol optimization and better characterization and standardization of the resultant cell types.

## 4. Human iPSC-Derived Astrocytes in Neurological Disorders

The classical drug discovery and validation method has some drawbacks, such as time-consuming procedures, the need for many animals for experiments, and the testing of a limited number of compounds under limited experimental conditions. Recent advancements in the available techniques led to high-throughput screenings, which can test hundreds or even thousands of compounds in less time and with lower relative costs. Although these procedures are more efficient than previous ones, they have poor validation rates due to the rare biological relevance of the screening platforms when related to the disease to treat and the use of non-human derivatives, which may not be related to specific human biology. The great advantage of using iPSC technology in developing treatments for human disease is evident. We will provide an overview of the relevance of iPSC-derived astrocytes in neurodegenerative diseases, including methods for differentiating disease-relevant cells, essential findings in drug development, and current trends for improving treatment.

### 4.1. Alzheimer’s Disease

AD is an age-related neurodegenerative disease that mainly affects memory and executive functions, which induce behavioral and neuropsychiatric changes. AD is characterized by the accumulation of amyloid-beta (Aβ) plaques and TAU-laden neurofibrillary tangles [72]. Almost 40 million people worldwide have dementia, most of them older than 60 years. This number is rapidly increasing and is estimated to double by at least 2050 [73], overtaking cancer as the second leading cause of death after cardiovascular diseases [74]. There is no treatment to stop the disease progression; therefore, identifying the exact cause and mechanism is most important. 

Astrocytes play an essential role in the pathology of AD. They are reported to be involved in various brain functions and are known to become reactive in different disease pathologies, changing their gene expression profile and metabolism [75]. Astrocytes maintain neuronal excitability and synaptic transmission by regulating ion concentrations. Astrocytes also constitute a significant source of cholesterol and other lipids critical for many cellular functions, as well as lipoproteins such as APOE, which are essential regulators of brain Aβ clearance and degradation [76]. Astrocytes of AD patients have shown increased GFAP expression and gamma-aminobutyric acid (GABA) production and release [77]. Furthermore, Aβ accumulates in AD astrocytes.

Several studies were performed using iPSC-derived neuronal and glial cells obtained from AD patients and healthy individuals. Most of these studies demonstrated that AD patients showed increased Aβ42 secretion [78]. Kondo and colleagues obtained iPSC-derived neurons and astrocytes from three AS patients, one with sporadic and two with familial AD, carrying the p.Glu693Asp or p.Val717Leu mutations of the amyloid precursor protein. They found an over-accumulation of Aβ oligomers in neurons and astrocytes obtained from AD iPSCs. They further evaluated the use of docosahexaenoic acid as a therapeutic agent to prevent symptoms [42]. 

In 2017, Oksanen and colleagues [79] generated iPSC-derived astrocytes from three AD patients with presenilin 1 (PSEN1) exon 9 deletion and found increased Aβ production and cytokine release and dysregulated Ca^2+^ homeostasis. They also showed increased oxidative stress and reduced lactate secretion and neuronal supportive function. Another study evaluated the reduction of morphological complexity and disturbing localization of marker proteins in iPSC-derived astrocytes obtained from AD patients with familial Alzheimer’s disease (FAD)-linked PSEN1M146L and sporadic Alzheimer’s disease (SAD)-linked APOE4 mutations [80]. iPSC-derived astrocytes from healthy individuals showed increased survival, maturation, and support of co-cultured human neurons; these features were diminished in astrocytes from AD patients [81]. Additionally, APOE4 cells displayed a higher accumulation of cholesterol and lower ability to internalize Aβ42 compared to isogenic APOE3 astrocytes. They also secreted less APOE protein, reducing lipidation [82].

These studies suggest that astrocytes are significantly implicated in AD pathology. Thus, iPSC-derived astrocytes from AD patients can be used as a testing platform for optimal pharmacological treatment individuation. Although informative, current studies employing iPSC-derived astrocytes from AD patients are still sparse. Additional studies are essential to clarify the role of astrocytes in AD further.

### 4.2. Parkinson’s Disease 

PD affects 7 to 10 million people worldwide, making it the second-most prevalent neurodegenerative disease after AD. Most PD cases are sporadic (85%), and only 15% of patients show familial mutations [83]. There are several hypotheses behind PD pathogenesis, including neuroinflammation, mitochondrial dysfunction, dysfunctional protein degradation, and alpha-synuclein (α-synuclein) pathology, but the exact cause is still unknown [84,85,86]. PD is mainly characterized by the loss of ventral midbrain dopaminergic neurons in the substantia nigra pars compacta and the presence of Lewy bodies in the brain. These pathological changes are considered responsible for the typical motor symptoms (bradykinesia, rigidity, rest tremor, and postural instability) seen in PD [87,88]. There is no proper cure for PD, and most interventions are aimed at relieving the motor symptoms with either dopamine replacement therapy or surgery [89].

In the early 1980s, researchers used fetal DA neural cells of human origin in a PD rat model for therapeutic purposes. They demonstrated that PD rats successfully restored functional activity after transplantation [90]. Ten years later, another group found a way to translate this approach to patients. They successfully transplanted human fetal DA neurons in a severe PD case, which engrafted and improved motor functions [91]. With the precedence of previous research, transplantation of iPSC-derived DA neurons is becoming a therapeutic opportunity for PD patients [68]. Some preclinical and clinical studies have used iPSC-derived DA neurons in PD rats and human models, demonstrating that neurons successfully improved motor functions and survival rate for at least two years [92,93]. In 2018, the so-called “Kyoto clinical trial” was started. The authors efficiently induced dopaminergic neurons from human iPSCs and purified dopaminergic progenitor cells by sorting. After in vivo preclinical studies in rats and monkeys, cell-based therapy for PD was started in humans [94]. At present, seven patients have been recruited, and the recruitment is now closed. The results of the study have not yet been published.

Apart from DA neurons, the contribution of glial cells to the pathogenesis of PD has been considered. A study compared the functionality of iPSC-derived astrocytes and dopaminergic neurons obtained from PD patients with the p.Gly2019Ser mutation and healthy controls. They found that astrocytes from PD patients secreted α-synuclein, which exerted a neurotoxic function on surrounding dopaminergic neurons, resulting in neuronal dysfunction. They also found dysfunctional chaperone-mediated autophagy and progressive α-synuclein accumulation in the PD iPSC-derived astrocytes [89].

Another study conducted by Sonninen and colleagues [95] used astrocytes derived from iPSCs of healthy donors and PD patients with p.Gly2019Ser and p.Asn370Ser LRRK2 and glucosylceramidase-beta mutations. The astrocytes from PD patients showed increased α-synuclein expression, which resulted in altered metabolism, disturbed Ca^2+^ homeostasis, and increased release of cytokines upon inflammatory stimulation. In addition, PD astrocytes also showed other signs of PD pathology, including increased levels of polyamines and polyamine precursors and decreased levels of lysophosphatidylethanolamine.

Transplantation is a growing field in PD involving iPSC-derived dopaminergic neurons. Further, the knowledge of the role of astrocytes is increasing. This occurrence can be essential in identifying pathological astrocytic phenotypes in human stem cell models of familial and sporadic PD.

### 4.3. Amyotrophic Lateral Sclerosis 

Amyotrophic lateral sclerosis (ALS) is a fatal neurodegenerative disease characterized by the loss of cortical and spinal cord motor neurons, causing muscle weakness, atrophy, and paralysis, with the death of patients by respiratory failure within 3–5 years after diagnosis [96]. ALS is a rare disease, affecting about 2–3 in 100,000 individuals, and approximately 90% of ALS cases are sporadic (sALS) due to multiple genetic, epigenetic, and environmental factors, while about 10% are familial (fALS), which is clinically indistinguishable from sALS [97]. Mutations in more than 20 genes are associated with fALS, the most important being mutations in the chromosome 9 open reading frame 72 protein (C9orf72), superoxide dismutase type 1 enzyme (SOD1), 43 kDa transactive response-DNA binding protein (TARDBP, TDP43), and fused in sarcoma/translocated in liposarcoma protein (FUS/TLS) [98,99]. ALS is a complex disease due to its multiple causes, also involving non-neuronal cells [100,101]. In particular, astrocytes have a detrimental role in ALS progression [102,103] with several mechanisms, including excitotoxicity, altered astrocyte metabolism, inflammation, and oxidative stress [104,105,106,107].

Several studies have shown the significant impact of iPSC-derived astrocytes bearing different ALS mutations. A study conducted by Serio and colleagues demonstrated that human iPSCs-derived astrocytes obtained from ALS patients with TDP-43 mutations significantly impaired the subcellular localization of TDP-43 and decreased cell survival ([43]; Table 1). Similarly, another study showed that differentiated astrocytes from human ESCs overexpressing the p.Gly93Ala SOD1 mutant secrete various factors toxic to spinal motor neurons [108]. The ephrin type-B1 receptor is upregulated in injured motor neurons and induces astrocytic signal transducer and activator of transcription 3 signaling, followed by a protective and anti-inflammatory signature in astrocytes. Tyzack and colleagues demonstrated that astrocytes obtained from iPSCs of ALS patients carrying the SOD1 mutation disrupted the EphB1 receptor and the downstream protective pathway [109].

As discussed above, it took almost 18 weeks to generate iPSCs from patient fibroblasts [8] and a further 6–8 weeks to differentiate the iPSCs into astrocytes [43,61]. An alternative in the field of cell reprogramming in ALS research came with a new protocol for converting fibroblasts into induced neural progenitor cells (iNPCs) while managing to bypass iPSCs, which lasts about 3–4 weeks ([54]; Table 1). In that study, Ferraiuolo and colleagues used skin biopsy samples obtained from sporadic ALS patients with either SOD1 mutations or C9orf72 repeat expansion. They determined that these astrocytes significantly reduced motor neuron survival in co-culture compared to controls [110].

ALS is a multi-factorial and multi-cellular disease where many genes play a decisive role, thus highlighting the limitations of animal models, each characterized by a specific gene mutation. Having access to samples from patients with different genetic signatures or sporadic is a significant advantage for the knowledge of pathology and stratifying the results based on individual patients.

### 4.4. Huntington’s Disease

HD is a hereditary, neurodegenerative disease caused by cytosine–adenine–guanine (CAG) repeat expansion in the huntingtin (HTT) gene, which codes for glutamine [111,112]. CAG repeats result in a polyglutamine tract, leading to progressive involuntary motor movements, cognitive disturbances, and dementia [113,114]. HD affects about 12 per 100,000 individuals in the European population [115]. The disease can occur from childhood to old age, with a mean age at onset of 45 years [116], and CAG extension strongly correlates with disease onset [117]. The HTT gene has critical roles in cellular homeostasis processes such as transcription, protein-protein interactions, transport, mitochondrial functions, cellular stress responses, and vesicular trafficking [118]. However, in pathological conditions, a mutation in the HTT gene by CAG expansion in exon 1 results in the expansion of polyglutamine residue at the N-terminus of the HTT protein [111]. A previous study also reported the formation of small oligomeric fragments and protein accumulation in the nucleus, leading to the death of medium spiny neurons (MSNs) in the striatum and other regions [118].

While neuron degeneration plays a critical role in the pathology of HD, glial cells are also involved. Dysfunctional astrocytes in the striatum lead to alteration of astrocytes structure and abnormalities of electrical properties, thereby maintaining ion homeostasis and neuronal signaling through the release of glutamate and adenosine triphosphate [119] and increasing pro-inflammatory and reducing anti-inflammatory cytokine production, as well as the brain-derived neurotrophic factor (BDNF) and Ccl5/RANTES chemokine [120]. 

Many studies have used HD mouse models to assess astrocyte functions. Diaz-Castro and colleagues performed RNA sequencing in two mouse models, the transgenic R6/2 mouse, likely reflecting juvenile-onset HD, and zQ175 knock-in mice (with over 175 CAG repeats) that developed slow, adult-onset HD. They performed experiments at three stages of the disease to sequence astrocyte-specific RNA. Striatal astrocytes showed altered transcriptional profiles at six months [37]. In the same mouse models, Tong and colleagues found a decrease in the functional expression of the inwardly rectifying Kir4.1 K+ channel, leading to a reduced capacity of extracellular K+ buffering [121]. 

A study showed that mutated HTT led to increased activation of the neuronal N-methyl-D-aspartate (NMDA) receptor, causing toxicity in the YAC128 mouse model of HD, expressing the full-length mutated HTT with an identical distribution of endogenous HTT, with 128 CAG repeats [122]. Two studies validated the inhibition of metabotropic glutamate receptor 5 (mGluR5), a positive regulator of astrocyte glutamate transporter function, using two mouse models (zQ175 and BACHD mice) that express mutated HTT containing 97 mixed GAA-CAG repeats with genetic ablation of mGluR5. The results indicated that mGluR5 regulates repressor element 1-silencing transcription factor/neuron-restrictive silencer factor (REST/NRSF) expression via the Wnt signaling pathway, highlighting the contribution of impaired REST/NRSF signaling to HD pathology [123,124]. 

In general, astrocytes reveal a multiplicity of alterations in HD mouse models. Several studies were conducted on iPSCs-derived astrocytes from human cells to model in vitro HD and reveal molecular disease mechanisms [49,111,112,125]. In 2019, Cho and colleagues designed a protocol to differentiate astrocytes from HD monkey iPSCs in 28 days [111] compared to previous protocols [51,126]. They demonstrated that iPSC-derived astrocytes had a lower expression of glutamate transporter 1, metabotropic glutamate receptor 1 or 5, and subunit 2 of the ionotropic glutamate receptor AMPA [111]. Furthermore, they recorded a loss of voltage-dependent K+ conductance and increased expressions of 60 kDa heat shock protein (HSP60), caspase 3 (CASP3), CASP9, and B-cell lymphoma 2 (BCL2), recapitulating cellular and molecular hallmarks of HD. They also showed increased expression of small-hairpin RNA against HTT (shHD) and reversed HD phenotypes in astrocytes. Astrocytes with elevated expression of shHD had significantly higher expression of peroxisome proliferator-activated receptor γ co-activator 1α and superoxide dismutase 2 compared to HD astrocytes but lower values compared to wild-type astrocytes [111].

A previous study conducted by Juopperi and colleagues revealed that human iPSC-derived astrocytes from HD patients exhibited autophagocytic vacuoles; still, they did not assess the functional maturity of these cells ([49], Table 1). Recently, researchers generated mature astrocytes from HD patient iPSCs, which showed Kir currents’ impairment, lengthened spontaneous Ca2+ waves, and reduced cell membrane capacitance [112]. 

From the studies in the literature, it appears that patient-derived astrocytes present apparent pathological features of the disease. Thus, iPSCs-derived astrocytes show promise as a valuable tool for HD modeling and emerge with great potential in approaches to cell replacement therapy.

### 4.5. Multiple Sclerosis 

MS, a chronic, auto-immune, demyelinating CNS disease, is the leading cause of neurological disability in young adults [127,128]. The symptoms start with acute episodes of neurological dysfunction, followed by a relapsing-remitting course (RRMS) [129]. Later, approximately 80% of patients develop secondary progressive MS leading to permanent disability, including limb weakness, sensory loss, vision disturbances, pain, and muscle spasms. The remaining patients directly reach the progressive phase, referred to as primary progressive MS [129,130]. The pathology of MS remains mostly unexplained; however, shreds of evidence suggest that activated immune cells from the periphery migrate to the CNS, forming lesions characterized by primary demyelination and relative preservation of axons [127]. Currently, available disease-modifying treatments are proven to suppress the inflammatory components in RRMS; however, these drugs are practically ineffective in progressive forms [128].

Reactive astrocytes are instrumental in forming MS plaques from the early stages of the disease, recruiting lymphocytes and damaging tissue, confining inflammation, and promoting lesion repair [127,131]. Reactive astrocytes contribute to the various neuroinflammatory responses, including the production of and reactivity to soluble mediators (e.g., cytokines and chemokines), but they also regulate oxidative stress and maintain BBB integrity and function [132]. Moreover, various studies suggest that reactive astrocytes are associated with a detrimental effect by exacerbating inflammation and inhibiting regeneration; meanwhile, they can also contribute to neuroprotection [127,131,132,133]. These results highlight the complex and dual role of astrocyte-mediated regulation during disease progression.

iPSC-derived astrocytes represent an attractive approach to enhancing our knowledge of astrocyte dysfunction contributing to MS etiology [18,127,134]. However, at present, only two studies have been conducted on iPSC-derived astrocytes assessing the risk variants leading to MS susceptibility, with contradictory conclusions [18,134]. Perriot and colleagues [18] reported no detectable differences between the profiles of pro-inflammatory cytokines secreted by astrocytes derived from relapsing-remitting MS patients and healthy controls. These results agree with the previous hypothesis that genetic risks rather than CNS cell population characteristics regulate the immune response [135]. On the other hand, Ponath and colleagues [134] suggested an effect of MS-linked genetic risk variant rs7665090GG on iPSCs-derived astrocytes from MS patients. They found that the risk variant was associated with increased NF-κB signaling and target gene expression, astrocyte and lymphocyte infiltration within MS lesions, and lesion extension.

Astrocytes can have different roles in MS pathology depending on the pathological stage, the damage mechanisms considered, and the patient. Studies in the literature have not yet elucidated the mechanisms that give astrocytes this dual role. Therefore, the iPSC-derived astrocyte model is a robust platform that can help evaluate these aspects and highligh therapeutic targets capable of modifying the harmful elements of astrocytes in MS or enhancing the positive ones.

### 4.6. Spinal Muscular Atrophy 

Spinal muscular atrophy (SMA) is an autonomous hereditary disease characterized by a loss of motor neurons leading to muscle atrophy, respiratory failure, and death [136,137,138]. SMA is a rare disorder with a frequency of 1:11,000 individuals; however, it is the leading cause of infant death [136,137,138]. SMA is mainly caused by deletion or mutations of the survival motor neuron 1 (SMN1) gene on chromosome 5q, which codes for the SMN protein, found in neuronal and non-neuronal cells, including astrocytes [138,139]. This protein is essential for normal development and functional homeostasis in all species. Humans carry a second, closely related gene, called SMN2, an SMN1 homolog that can also produce the SMN protein. The greater the number of copies of this gene, the milder the SMA [140]. Currently, there are only four approved drugs for SMA: Nusinersen, onasemnogene, abeparvovec, and risdiplam; however, their long-term benefits remain to be assessed [141]. 

Astrocytes are crucial components contributing to MN damage and loss in SMA [136,138,142]. Restoring SMN selectively in astrocytes using scAAV-SMNgfap viral delivery, Rindt and colleagues [138] observed improved neuromuscular circuitry and an increase in the number and lifespan of neuromuscular junctions. They also hypothesized that astrocyte functions are disrupted in SMA, measured by the increased number of GFAP-positive cells and elevated production of pro-inflammatory cytokines. This increased expression of inflammatory cytokines was partially normalized in treated mice, suggesting that astrocytes contribute to the pathogenesis of SMA [138].

Many studies that obtained astrocytes from patient-derived iPSCs were published in the last 10 years, investigating the molecular mechanism of the astrocyte function in SMA. One study showed that iPSC-derived astrocytes from SMA1 patients presented various dysfunction, including morphological changes, up-regulation of GFAP protein, disrupted Ca^2+^ signaling, increased extracellular signal-regulated kinase 1–2 activation, and decreased glial cell-derived neurotrophic factor (GDNF) production [139]. Another study demonstrated increased production and secretion of nuclear factor κB and miR-146a in SMA iPSC-derived astrocytes and up-regulation of GATA binding protein 6 gene transcripts and proteins correlated with SMA severity [142]. A further study observed that SMA iPSC-derived astrocytes altered neighboring cell morphology and increased reactive oxygen species and catalase levels. In contrast, re-expression of SMN prevented astrocyte reactivity and restored astrocyte function. Mitochondrial bioenergetics and oxidative stress markers were unchanged, suggesting that oxidative stress genes and antioxidant defenses are not activated in the iPSC system [136]. 

The literature supports the role of astrocytes in SMA and indicates that deletion of the SMN protein strongly influences astrocyte functions. Therefore, using astrocytes obtained from iPSCs derived from SMA patients may represent a critical cell model to investigate SMA cellular mechanisms and test other potential drugs in the future. 

## 5. Conclusions

Astrocytes are actively involved in brain functions and the initiation and progression of neurodegenerative diseases. Human astrocytes are more complex than their rodent counterpart; therefore, a better understanding of the role of astrocytes in human pathologies is needed. The discovery and development of human iPSC technologies represent a novel and reliable platform for neurodegenerative disease modeling in humans. iPSCs-derived astrocytes with disease-specific gene mutations may represent a powerful tool for understanding the cellular and molecular disease mechanisms in vitro. These models could clarify astrocytes’ contribution to neurological and neurogenerative disorders, including the identification of new therapies that can counteract the disease course of these devastating pathologies.

## Figures and Tables

**Figure 1 biomolecules-12-00344-f001:**
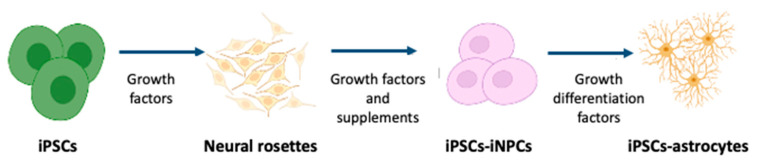
Schematic steps for differentiation of iPSC to astrocytes.

**Figure 2 biomolecules-12-00344-f002:**
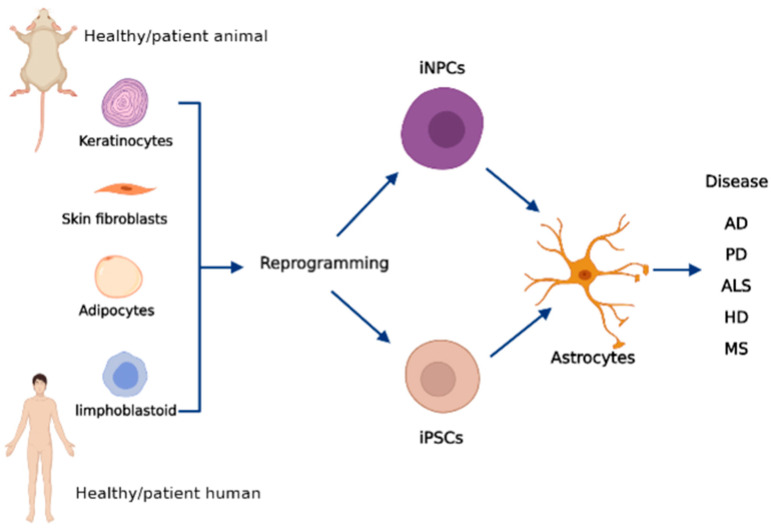
Schematic representation of the preparation of induced pluripotent stem cells (iPSCs) and induced neural progenitor cells (iNPCs) from human and rodent somatic cells and their transformation into astrocytes relevant to various neurodegenerative disorders.

**Table 1 biomolecules-12-00344-t001:** Protocols for the differentiation of human iPSCs into astrocytes.

Reference	Cell Source	Key Players	Research Outcome
Hue et al., 2010 [46]	Human iPSCs	RA (100 nM) SHH: 100 ng/mL cAMP: 1 µM	GFAP+ cells after 3 months
Krencik., 2011 [47]	Human iPSCs	RA: 0.5 µM FGF8: 50 ng/mL SHH: 500 ng/mL EGF and FGF2: 10 ng/mL CNTF: 10 ng/mL LIF: 10 ng/mL	Uniform population of mature astrocytes
Emdad et al., 2012 [48]	Human iPSCs	SB43152:10 µM Noggin: 500 ng/mL	50–70% of GFAP+ cells at week 5
Juopperi et al., 2012 [49]	Human iPSCs	bFGF: 20 ng/mL	GFAP+ cells after 2–3 months
Lafaille et al., 2012 [50]	Human iPSCs	EGF/FGF2: 20 ng/mL Sonic C2511: 125 ng/mL FGF8: 100 ng/mL BDNF: 20 ng/mL Ascorbic acid: 0.2 mM	GFAP+ cells after 90 days
Serio et al., 2013 [43]	Human iPSCs	EGF/FGF2: 20 ng/mL CNTF: 10 µg/mL	GFAP+ cells after 8 weeks
Shaltouki et al., 2013 [51]	Human iPSCs	bFGF: 20 ng/mL CNTF: 5 ng/mL BMP: 10 ng/mL bFGF: 8 ng/mL Activin A: 10 ng/mL Heragulin 1β: 10 ng/mL IGFI: 200 ng/mL	GFAP+ cells after 5 weeks
Sareen et al., 2014 [52]	Human iPSCs	EGF: 100 ng/mL FGF2: 100 ng/mL Heparin: 5µg/mL RA: 0.5 µM	Increased GFAP+ cells
Mormone et al., 2014 [53]	Human iPSCs	FGF2: 10 ng/mL EGF: 20 ng/mL Noggin: 500 ng/mL FGF+EGF+CNTF: 20 ng/mL	GFAP+ cells after 28–35 days
Caiazzo et al., 2014 [44]	Human Fibroblast	NFIA: 13.2% NFIB: 16.1% SOX9: 13.2%	GFAP+ cells in 2 weeks
Meyer et al., 2014 [54]	Human Fibroblast	KLF4, OCT-3/4, SOX2, c-MYC1	GFAP+ cells in 18 days
Zhou et al., 2016 [55]	Human iPSCs	LDN193189: 0.2µM SB431542: 10µM AA: 0.2 mM	GFAP+ cells in 4 weeks
Canals et al., 2018 [56]	Human iPSCs	NFIB SOX9	GFAP+ cells in 14 days
Tchieu et al., 2019 [57]	Human iPSCs	NFIA	GFAP+ cells in 5 days
Gatto et al., 2021 [58]	Human iNPCs	FBS: 10% Pen-Step: 1% N2: 0.2%	GFAP+ cells in 7 days

Abbreviations. RA: Retenoic acid; AA: Ascorbic acid; SHH: Sonic HedgeHog; FGF: Fibroblast growth factor; EGF: Epidermal growth factor; CNTF: Ciliary neurotrophic factor; BMP: Bone morphogenetic proteins; LIF: Leukemia inhibitory factor; cAMP: Cyclic AMP; NFIA: Nuclear factor IA; NFIB: Nuclear factor IB; SOX9: SRY-Box Transcription Factor 9; Pen-Strep: Penicillin-Streptomycin; Klf4: Krüppel-like factor 4; Oct 3/4: Octamer-binding transcription factor 3/4; SOX2: Sex determining region Y; c-MYC1: Master Regulator of Cell Cycle Entry and Proliferative Metabolism.

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
