# Peer review of "Insights into Human-Induced Pluripotent Stem Cell-Derived Astrocytes in Neurodegenerative Disorders"

_biomolecules, 2022, doi:10.3390/biom12030344_

Round 1

Reviewer 1 Report

In this review, authors discuss the developed protocols and the applications of human iPSC-derived astrocytes in several neurodegenerative conditions. Although this manuscript could be of interest for the journal, several modifications must be performed before its acceptance for publication.

Major changes:

*Lines 34-48: this section is a bit disorganized and needs to be rearranged i.e. the discovery of iPSC technology must be described before its applications.

*Line 44: describe the advantages of using induced pluripotent stem cells versus animal models.

*The section named “Human iPSCs-Derived Astrocyte Preparation” is very relevant for the review. However, it is written in a very confused and repetitive way. For example, what the authors mean by the sentence: “…to modify human somatic cells to almost embryonic pluripotent undifferentiated cells”?. The description of the iPS discovery is repeated again, etc. This section should be written in a much more clear way and the inclusion of a table summarizing the protocols to differentiate iPSCs into astrocytes would be much appreciated.

*Lines 148-150: In this sentence “However, most of the studies used the combination of morphogens and growth factors such as Retinoic acid, wingless-type MMTV integration site protein family, fibroblast growth factor, and growth differentiation factor in differentiating astrocytes from” there is something missing.

*Lines 170-172: This sentence “Further development of the technology allowed the use iPSCs from well-established models for developmental and adult-onset diseases, in the form of either two-dimensional (2D) cell cultures or three-dimensional (3D) organoids [9, 52]” is not clear. Please, write it again.

*Lines 181-200: This paragraph is also not clear. Thus I advise the authors to write it again.

*In the section of Parkinson disease the Kyoto trial must be included (Takahashi J. Regen Ther. 2020).

*Lines 428-441: Please revise this paragraph because the end does not fit well with the rest.

*Mutations nomenclature is incorrect. Please revise throughout the manuscript and follow the current recommendations (https://varnomen.hgvs.org/).

Minor points:

*Line 33: …high severity of their symptoms and lack OF effective therapies

*Line 50: Abbreviation for MS must be described. One possibility could be in lines 31-32.

*Legend of figure 1: there is an extra full stop

*Legend of figure 3 is not clear and should be rewritten.

*There are several abbreviations in text that must be explained when written for the first time.

*English must be revised by a native English speaker.

Reviewer 2 Report

In this work, Dr. Kumar and colleagues review the use of hiPSC-derived astrocytes in the research on the most common neurodegenerative disorders, i.e., Alzheimer’s disease, Parkinson’s disease, amyotrophic lateral sclerosis, Huntington’s disease, multiple sclerosis, and spinal muscular atrophy.

The use of hiPSCs has had a huge impact on biomedical research, especially in the neuroscience field, as human neurons and glial cells are not accessible. Astrocytes in particular have taken the center of the stage as they play an important role in both neurodegenerative and cognitive pathologies. This review therefore touches upon a number of subjects that are of high interest in today’s neuroscience research. I believe this work should be published; however, I found some sections not easy to follow and I recommend some changes as detailed below.

Section 2. Astrocytes Phenotypes and Functions. Here the authors comment on the A1 / A2 subdivision of reactive astrocytes as ‘neurotoxic’ and ‘neuroprotective’, respectively, as described in several papers from the Barres group. While this classification has had a wide success and has encountered immediate acceptance from the scientific audience, recently a Consensus Statement has been published from several scientists, calling for a more cautious use of such classification, since the types and roles of activated astrocytes are likely to be much more diversified and subtle (Escartin et al, Nat Neurosci 2021). Thus, I would recommend adding to the discussion some considerations on this point.

Section 3. Human iPSCs-Derived Astrocyte Preparation. I found this part very interesting, but the information is presented in a way that lacks a logic structure and is therefore confusing. For example, the authors talk about protocols for both neurons and astrocytes, and it is not always clear what cell type they are referring to. I would recommend focusing exclusively on protocols to induce astrocytes – unless mentioning neurons is important to understand the methods – and to follow a chronological order so that the reader can really follow the development of better protocols over the years. Also, I would appreciate that pros and cons are stated for all the approaches. Last, I would dedicate a conclusive paragraph to the protocols that are most commonly used today, commenting on their efficacy and on the parts that are instead still to be optimized.

Section 4. Human iPSCs-Derived Astrocytes in Different Neurological Disorders. I enjoyed reading this part as it is well organized and clear. The only thing that in my opinion is lacking is a more personal opinion from the authors about the main achievements obtained through the use of hiPSCs in the various pathologies. For each pathology the results from several papers are described, but it would be good to have a conclusive comment to convey the state-of-the-art of this technology for the various diseases.

Reviewer 3 Report

The authors here provide a review of iPSC derived astrocytes, from improvements in the experimental techniques to the study of disease mechanisms, followed by a review of some specific diseases from work in which iPSC derived astrocytes was used. The review could potentially be of use to the field, but a complete overhaul would be needed before the manuscript could be considered for publication. A list of the major concerns follows.

   1. In general, the manuscript reads like a rough draft. The writing could be greatly improved both in terms of the English grammar and style as well as the overall organization. Often overgeneralizations are made with a lack of specifics or supporting information, leaving the work superficial and of limited usefulness. For one example, line 72 astrocytes release “harmful molecules”. This is completely useless information at best; may lead, misleading. At other times there are long run-on sentences (more apparent in the disease-specific section) which makes the writing uninterpretable. An example is lines 359-364. Clearer, more concise writing with specific supporting information from the works is needed throughout.
    2. Unfortunately, a review is only as good as its citations. This is a glaring weakness of the manuscript, greatly limiting its usefulness. There are very often “empty statements” where a sentence provides some information only to be left uncited. For example, the very first sentence. But this is a common theme throughout. Other times, the review mis-cites papers (cites papers that focus on something else, leaving the sentence uncited). In other cases, citations of other reviews are made, rather than citing the primary literature. Citing reviews should only be done sparingly, and this needs to be clearly marked “reviewed previously by so-and-so”. For an example of the above, lines 93 - 97 mention reactive astrocytes in neurodevelopmental disorders, then cite a review on reactive astrocytes and another paper on the effects of aging on reactive astrocytes. No primary literature on reactive astrocytes in neurodevelopmental disorders is provided. Another poorly cited section is the paragraph between lines 106-111. None of it is cited, it is overgeneralized, and it lacks specifics. Because of this, removing it entirely would improve the manuscript since it provides no useful information. Unfortunately, this is a microcosm of the manuscript.
    3. The area of reactive astrocytes is a quickly moving field. The section on reactive astrocyte subtypes and heterogeneity is already outdated. The field has moved quickly past the A1 vs. A2 classification. The “neurotoxic” reactive astrocyte type is widely acknowledged as problematic and a great oversimplification. There is little evidence that neurotoxins are secreted. Rather, astrocytes lose their supportive functions, which, because they are tuned to controlling excitability, contribute to neuronal hyperexcitability and eventually synaptic loss and excitotoxicity. The authors are directed to review the latest literature on this topic and update this section accordingly.
    4. Because the writing lacks clarity and does not provide specific information, it is a struggle as a reader to know if the review is focusing on work done in culture, or iPSC-derived astrocytes that are then reintroduced into a mouse disease model, or both. It appears that most of the work is characterizing features of astrocytes or astrocytes co-cultured with neurons in vitro, but this is not clear. In each case the experimental model used should be clearly described, with the results and conclusions critically evaluated.
    5. Because the format is to enumerate citations rather than using the author-date method, the section on improvement in cell culture techniques should be reorganized chronologically and include the dates. Otherwise it lacks organization and is difficult to follow. It is kind of all over the place.
    6. It is not always clear if the authors are referring to astrocytes or neurons. Sometimes “neural” is used, which could be either or both cell types.
    7. The whole disease section provides superficial information, often a single sentence per paper, without any specific information on mechanisms. There are empty sentences that provide no or useless information, like “Astrocytes contribute to the neuroinflammatory response, including neurotoxicity and neuroprotection, by producing and releasing pro-and anti-inflammatory factors”, or that astrocytes release neurotoxins. This kind of defeats the whole purpose of a review paper. 

Round 2

Reviewer 1 Report

Manuscript has been improved but there are minor points that should be correct:

1. Line 254: Further development of the technology allowed the use OF iPSCs in combination...

2. Mutations nomenclature is still incorrect. Please revise throughout the manuscript and follow the current recommendations:  (https://varnomen.hgvs.org/). 

Reviewer 3 Report

The authors did address most of the initial concerns. Especially the section on reactive astrocytes and the addition of the table in chronological order of the developments in methodology for iPSC derived astrocytes are notable improvements. The authors could consider referencing their own table in later sections of the manuscript that describe an advancement in technique, including sections in the study of a particular disease, for example lines 476-483 on ALS describing an advancement in protocol by Ferraiuolo et al. The disease specific sections went largely unchanged and could still benefit from providing more mechanistic detail. Last, despite the proofreading, the MS is still a little sloppy with some misspellings and grammar issues. For example line 321 "those" is misspelled and line 272 "obtaining" is misspelled. The MS would benefit from additional proofreading. Including some of these minor suggestions would improve the work without need for re-review.
